# Brain Re-Irradiation Or Chemotherapy: a phase II randomised trial of re-irradiation and chemotherapy in patients with recurrent glioblastoma (BRIOChe) – protocol for a multi-centre open-label randomised trial

Eleanor M Hudson [1], Samantha Noutch,[1] Joanne Webster,[1] Sarah R Brown [1], Florien W Boele,[2,3] Omar Al-Salihi,[4] Helen Baines,[5] Helen Bulbeck,[6] Stuart Currie,[7] Sharon Fernandez,[2] Jane Hughes,[2] John Lilley,[8] Alexandra Smith,[1] Catherine Parbutt,[9] Finbar Slevin,[2,10] Susan Short,[2,10] David Sebag-Montefiore,[2] Louise Murray[2,10]

For numbered affiliations see end of article.

**Correspondence to**
Dr Louise Murray;
L.J.Murray@leeds.ac.uk

## ABSTRACT

**Introduction** Glioblastoma (GBM) is the most common adult primary malignant brain tumour. The condition is incurable and, despite aggressive treatment at first presentation, almost all tumours recur after a median of 7 months. The aim of treatment at recurrence is to prolong survival and maintain health-related quality of life (HRQoL). Chemotherapy is typically employed for recurrent GBM, often using nitrosourea-based regimens. However, efficacy is limited, with reported median survivals between 5 and 9 months from recurrence. Although less commonly used in the UK, there is growing evidence that re-irradiation may produce survival outcomes at least similar to nitrosourea-based chemotherapy. However, there remains uncertainty as to the optimum approach and there is a paucity of available data, especially with regards to HRQoL. Brain Re-Irradiation Or Chemotherapy (BRIOChe) aims to assess re-irradiation, as an acceptable treatment option for recurrent IDH-wild-type GBM.

**Methods and analysis** BRIOChe is a phase II, multi-centre, open-label, randomised trial in patients with recurrent GBM. The trial uses Sargent's three-outcome design and will recruit approximately 55 participants from 10 to 15 UK radiotherapy sites, allocated (2:1) to receive re-irradiation (35 Gy in 10 daily fractions) or nitrosourea-based chemotherapy (up to six, 6-weekly cycles). The primary endpoint is overall survival rate for re-irradiation patients at 9 months. There will be no formal statistical comparison between treatment arms for the decision-making primary analysis. The chemotherapy arm will be used for calibration purposes, to collect concurrent data to aid interpretation of results. Secondary outcomes include HRQoL, dexamethasone requirement, anti-epileptic drug requirement, radiological response, treatment compliance, acute and late toxicities, progression-free survival.

**Ethics and dissemination** BRIOChe obtained ethical approval from Office for Research Ethics Committees Northern Ireland (reference no. 20/NI/0070). Final trial results will be published in peer-reviewed journals and adhere to the ICMJE guidelines.

**Trial registration number** ISRCTN60524.

## STRENGTHS AND LIMITATIONS OF THIS STUDY

⇒ Brain Re-Irradiation Or Chemotherapy (BRIOChe) is multi-centre randomised trial exploring re-irradiation and chemotherapy for recurrent glioblastoma (GBM).

⇒ The trial will collect much needed survival and health-related quality of life data for patients with recurrent GBM treated with re-irradiation.

⇒ BRIOChe focuses on the underserved population of recurrent GBM patients, for whom there is a significant lack of treatment options and UK-based trials.

⇒ The trial design and decision criteria are based on available historical data at the time of development, where patient populations may differ from those in the BRIOChe trial.

⇒ The trial does not provide formal comparative evidence between treatment arms. Rather the parallel concurrent control arm provides essential randomised data for benchmarking and interpretation.

## INTRODUCTION

### Glioblastoma

Glioblastoma (GBM) is the most common adult primary malignant brain tumour, with an incidence of 5/100 000 per year in England. Prognosis is poor, with an average survival of 12–15 months.[1]

GBM is incurable, with treatment from the point of diagnosis aiming to improve survival,

delay disease progression and manage symptoms/ preserve health-related quality of life (HRQoL). Optimal treatment at first presentation is aggressive, including debulking surgery (if feasible), chemo-radiotherapy and adjuvant chemotherapy. However, almost all patients will relapse within the brain, with a median progression-free survival (PFS) of approximately 7 months.[1]

The burden placed on patients, their informal caregivers and the healthcare system as a result of neurological disability and the care requirements for patients with brain tumours is significant.[2–4] GBM is often accompanied by debilitating physical and neurological symptoms, such as paresis, sensory loss, seizures, fatigue, mood issues and cognitive deficits.[5–7] This symptom burden can affect patients' ability to function independently and negatively impact on HRQoL for both patients and their informal caregivers.

### Recurrent glioblastoma

The optimal treatment for recurrent GBM is undefined and treatment decisions are made on a case-by-case basis, based on the site, volume and time of recurrence in relation to the original diagnosis, with consideration for patient age, co-morbidities and performance status.

In the UK, chemotherapy is commonly employed in the setting of recurrent GBM, typically using nitrosourea-based regimens. These DNA alkylating agents are highly lipophilic, and are therefore able to penetrate the blood-brain barrier, making them suitable for the treatment of brain tumours. Nitrosourea-combination regimens have not proven superior to single agent nitrosoureas, resulting in variations in practice as to which regimens are employed.[8] Efficacy is limited, with reported median overall survivals of between 5 and 9 months from recurrence.[9–17] Side effects may include fatigue, nausea, vomiting, anorexia, constipation, paraesthesia, rash and myelosuppression. There is currently a relative lack of HRQoL data for patients receiving nitrosourea-based chemotherapy for recurrent GBM,[13 16 18] making it difficult to determine whether the benefits of chemotherapy outweigh its burdens.

Surgery is another treatment option at point of recurrence, however, it is usually reserved for a minority of excellent performance patients with well-localised disease, although high-level evidence to support its role is lacking.[9 19–21]

### Re-irradiation for recurrent glioblastoma

Re-irradiation is a further treatment option for recurrent GBM, although at present it is not commonly used in the UK. A meta-analysis of re-irradiation for recurrent GBM[22] included 50 studies of re-irradiation, of which 35 employed external beam radiotherapy (EBRT); the remainder used brachytherapy. Of the patients who received EBRT, 6-month and 12-month overall survival (OS) was 70% and 34%, respectively.[22] PFS was 40% and 16% at 6 and 12 months, respectively. Grade 3+ toxicity was reported in 7% of patients. Overall, only 13 of the studies were

of prospective design and only 9 were considered by the authors to be of good quality. Similarly, a systematic review of re-irradiation in GBM presented comparable outcomes based on a pooled, population-weighted analysis, where mean adjusted OS following hypofractionated EBRT was 10.1 months (95% CI 9.7 to 10.5 months) and the rate of radionecrosis was 7.1% (95% CI 6.6% to 7.7%).[23] Despite the promising outcomes reported after re-irradiation, the poor quality of existing evidence highlights the urgent need for a high-quality, prospective, randomised assessment of re-irradiation, including HRQoL assessment.[23 24] More recently, the phase II RTOG1205 trial, which randomised patients with recurrent GBM between bevacizumab (not routinely available in the UK) and bevacizumab plus re-irradiation 35 Gy in 10 fractions, demonstrated a median OS of 10.1 months in the combination arm.[25] There was no difference in OS between arms, the primary endpoint of the trial, however, PFS was improved compared with bevacizumab alone.[25] Treatment was also well tolerated, with 5% grade 3+ acute toxicities and no delayed high-grade adverse events reported.

### Rationale for study

While the limited existing evidence suggests that re-irradiation has a potential positive impact on OS and HRQoL, to date, no trial has randomised patients with recurrent GBM between re-irradiation and chemotherapy. Addressing this research gap is a top priority for patients, informal caregivers and healthcare professionals.[26]

The Brain Re-Irradiation Or Chemotherapy (BRIOChe) phase II prospective trial will randomise patients with recurrent GBM to re-irradiation (35 Gy in 10 fractions, a commonly adopted re-irradiation regimen[27–32]), or nitrosourea-based chemotherapy. The aims will be to explore the efficacy and HRQoL impact of re-irradiation. This study will collect concurrent data for patients receiving chemotherapy for validation ('benchmarking') of outcomes. The study is not designed to statistically compare the two arms for the primary outcome of survival, which would require a larger number of patients. By establishing the feasibility of randomisation and providing prospective controlled data on survival and HRQoL impacts associated with re-irradiation, the BRIOChe trial aims to demonstrate re-irradiation as a possible alternative treatment option for patients with recurrent GBM and inform the design of future research.

## METHODS AND ANALYSIS
### Design and aim

BRIOChe is a phase II, multi-centre, open-label, randomised trial in patients with recurrent GBM. The aim of the study is to assess re-irradiation as an alternative treatment to chemotherapy in this group of patients. BRIOChe uses a Sargent's three-outcome, single-stage, single-arm design, with the inclusion of a chemotherapy calibration arm for concurrent data for benchmarking. Following an amendment due to slow

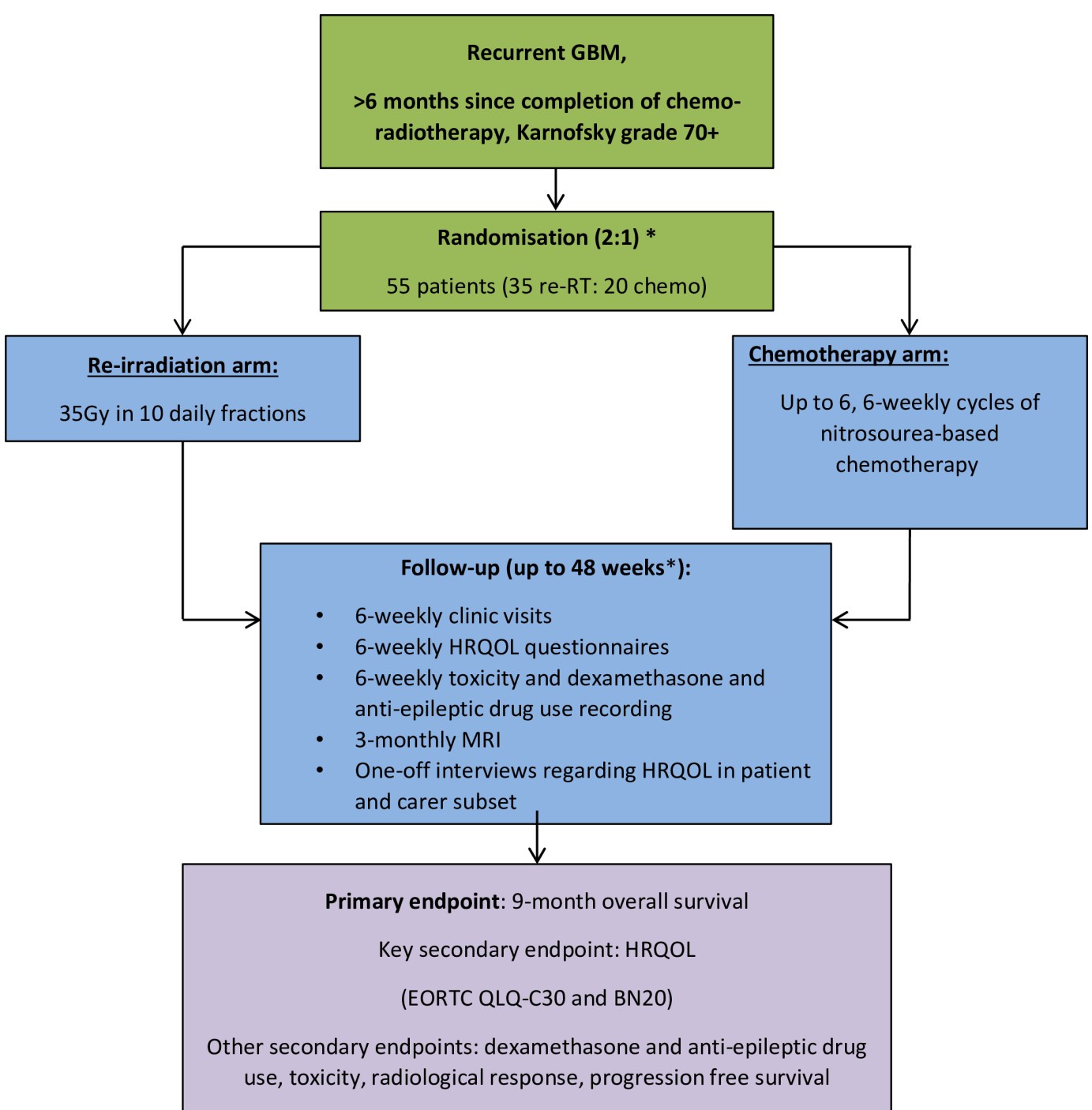

**Figure 1** Brain Re-Irradiation Or Chemotherapy trial schema. *Allocation ratio originally 1:1, ratio was adapted to 2:1 during the recruitment phase. EORTC QLQ-C30 and BN20, European Organisation for Research and Treatment of Cancer Quality of life questionnaire core 30 and Brain Cancer module 20; GBM, glioblastoma; HRQoL, health-related quality of life.

recruitment, the trial will now recruit approximately 55 participants from 10 to 15 UK radiotherapy sites. The original sample size was 70 based on a 1:1 allocation ratio, to receive either re-irradiation or nitrosourea-based chemotherapy (lomustine+/−procarbazine+/−vincristine). The allocation ratio was adapted during recruitment to 2:1, further details are presented in the sample size section.

The trial schema is presented in figure 1.

### Trial objectives

The primary objective of BRIOChe is to assess the proportion of participants alive at 9 months post-start of treatment in the re-irradiation arm.

The key secondary objective of BRIOChe is to evaluate the effect of re-irradiation on HRQoL. Questionnaires to be completed by participants include the European Organisation for Research and Treatment of Cancer (EORTC) Quality of life questionnaire core 30

**Table 1** Summary of key inclusion and exclusion criteria

| Inclusion criteria | Exclusion criteria |
|---|---|
| Histologically proven diagnosis of GBM, based on original pathology | Pregnant (positive pregnancy test) or lactating |
| First recurrence of GBM, with contrast enhancing disease, following primary treatment | Critical normal brain structures treated above usual tolerance during initial radiotherapy |
| Agreement of a consultant neuro-radiologist that imaging changes are in keeping with recurrence and not pseudo-progression | Recurrence with leptomeningeal disease or only leptomeningeal disease |
| ≥6 months since completion of primary radiotherapy | More than three enhancing lesions present on MRI or multi-focal recurrence |
| History of standard dose, conventionally fractionated CNS radiotherapy | IDH1/2 mutant tumours on original pathology |
| As a minimum, patients will have completed at least 2 weeks of temozolomide, concurrent with their original radiotherapy | GBM with known features of PXA, BRAF mutations or 1p19q codeletion (on original pathology or updated pathology if available) |
| Up to and including three enhancing lesions (size and volume criteria included in online supplemental file 1) | Prior invasive malignancy (except non-melanomatous skin cancer), unless disease free for a minimum of 1 year |
| Karnofsky performance status 70+ | Severe active comorbidity making patient unsuitable for chemotherapy or re-irradiation |
| Adequate haematological, renal and hepatic function | Prior allergic reaction to nitrosoureas |
| No contra-indication to lomustine | Any recognised genetic syndromes causing sensitivity to radiotherapy |
| Patients must be able to provide study-specific informed consent | Contra-indication to MRI or gadolinium |
| Age 18 or over | Previous radiotherapy dose distribution unavailable |
| | Previous systemic therapy or re-irradiation for recurrent GBM |

CNS, central nervous system; GBM, glioblastoma.

(QLQ-C30) and Brain Cancer module (BN20). HRQoL will also be explored in greater detail through one-off semi-structured qualitative interviews in a subsample of patients and their informal caregivers.

Additional secondary objectives include evaluating:
▶ Dexamethasone requirement.
▶ Anti-epileptic drug requirement.
▶ Radiological response in accordance with Response Assessment in Neuro-Oncology (RANO) criteria.
▶ Treatment compliance.
▶ Acute (measured from randomisation to 12 weeks post end of treatment) and late toxicities (after 12 weeks post end of treatment) as per Common Terminology Criteria for Adverse Events (CTCAE V.5.0); radionecrosis is a toxicity of particular interest.
▶ PFS.
▶ OS.

## Study population

All inclusion/exclusion criteria must be met and written informed consent obtained prior to recruitment (key criteria summarised in table 1); full criteria and a sample patient consent form are included as online supplemental files 1 and 2, respectively. The formal assessment of eligibility and informed consent may only be obtained by the principal investigator or an appropriate medically qualified doctor. Patients will be screened from clinic lists. Disease eligibility will be based on the diagnostic MRI scan where recurrence is identified. For patients who are randomised to re-irradiation, if significant progression is observed on the re-irradiation planning imaging (with Gross Tumour Volume (GTV) >75 cm$^3$ or doubling compared with the baseline MRI), such cases will be reviewed with the Radiotherapy Trials Quality Assurance (RTTQA) group before proceeding with re-irradiation. Patients who are randomised to chemotherapy who remain clinically stable would not routinely receive updated imaging prior to commencement of chemotherapy but in these cases the interval to starting chemotherapy is anticipated to be shorter than that to starting re-irradiation, providing less time for significant progression.

## Randomisation and recruitment

Randomisation will be performed centrally using University of Leeds Clinical Trials Research Unit (CTRU) automated 24-hour randomisation system. A computer-generated minimisation programme incorporating a random element will be used to ensure the treatment groups are well balanced for the following factors:
▶ Age (<50 or ≥50 years).

- ► Time to randomisation from completion of previous radiotherapy (≤12 months or >12 months).
- ► $O^6$-methylguanine-DNA-methyltransferase (MGMT) promoter methylation status (methylated, unmethylated or unknown).
- ► Randomising site.

The intended recruitment period is 2 years (BRIOChe opened to recruitment on 22 June 2021).

## Sample size

Thirty-three patients are required in the re-irradiation arm, 35 allowing for 5% drop out. With a 1:1 randomisation allocation ratio to include a calibration arm, the original sample size was 70 patients in total. As a result of slow recruitment, the randomisation allocation ratio was adapted in a protocol amendment to 2:1 (re-irradiation: chemotherapy), to allow a higher proportion of recruited patients into the intervention arm. The subsequent sample size was estimated to be 55 patients in total, with 35 in the re-irradiation arm and approximately 20 in chemotherapy arm.

This study is powered to demonstrate that the treatment strategy of re-irradiation demonstrates sufficient efficacy to warrant further large-scale evaluation, based on 9-month survival rates.

A Sargent's three-outcome, phase II, single-stage, single-arm design will be used to determine whether re-irradiation demonstrates sufficient efficacy to warrant further evaluation.[33] The trial is designed to test the null hypothesis (H0) that the proportion of participants alive at 9 months is <25% where re-irradiation would not be deemed worthy of further investigation, against an alternative hypothesis H1 of >45% where re-irradiation would be deemed worthy of further investigation. If the proportion of participants alive at 9 months is ≥25% and ≤45%, this is an inconclusive region where neither the null or alternative hypothesis would be rejected and the decision regarding further investigation would be based on other factors. The following operating characteristics are used: type I error=0.05, type II error=0.06, prob (correctly rejecting the H1) = 0.82 (eta), prob (correctly rejecting the H0)=0.79 (pi).

## Treatment regimen
### Re-irradiation arm

The radiotherapy prescription dose will be 35 Gy in 10 fractions, a commonly employed schedule,[27–32] delivered daily Monday-Friday over 2 weeks. Treatment will be delivered with intensity modulated radiotherapy (IMRT)/volumetric modulated arc therapy or tomotherapy (VMAT). Optimally, normal tissue constraints will be based on a 'dose remaining' approach, whereby an original organ at risk (OAR) dose (allowing for 25% repair) is subtracted from a cumulative tolerance to give a 'dose remaining' for re-irradiation. To save clinicians time and reduce errors in performing such calculations, the radiotherapy guidelines contain tables of 'OAR doses remaining' for ease of reference. Where these optimal constraints cannot be

achieved, mandatory constraints are those used in the RTOG1205 trial, which also delivered a dose of 35 Gy in 10 fractions.[31] Treatment compliance data to radiotherapy, including fractions, doses, interruptions and reasons will be collected weekly during treatment.

### Chemotherapy arm

As per usual UK practice in the setting of relapsed GBM where the interval to recurrence is insufficiently long to use re-challenge temozolomide and/or when patients have MGMT unmethylated tumours, chemotherapy will be lomustine-based. These drugs have been used in this setting for many years. A variety of lomustine-based regimens are used across the UK including single agent lomustine, PCV (procarbazine, lomustine and vincristine), PC (procarbazine and lomustine) or CV (lomustine and vincristine). Lomustine-based regimens will be delivered as per each centre's standard practice. Doses will be recorded at each cycle and (based on a survey of participating centres regarding their usual practice) it is recommend that these will consist of:

- ► Lomustine 100–130 mg/m$^2$, day 1, PO, on a 42day cycle

  +/−
- ► Procarbazine 50–100 mg/m$^2$, once daily on days 1–10 or days 2–11, PO, on a 42day cycle

  +/−
- ► Vincristine 1.4–1.5 mg/m$^2$, IV day 1, on a 42day cycle. (Dose capping is as per the institution's usual practice.)

Treatment compliance data to chemotherapy will be collected from patient diaries. Dose modifications and reasons for changes will also be collected. Suggested dose modifications to chemotherapy based on haematological, renal and hepatic functioning are provided in the study protocol.

On further disease progression, patients may receive the alternative treatment according to local policy.

### Trial assessments and follow-up

Full eligibility criteria (table 1, online supplemental file 1), will be assessed and confirmed prior to randomisation. Further assessments will be conducted at multiple timepoints. An overview of data collection can be seen in table 2. Data will be collected from all participants, until progression or death. Where available, date of death will be collected for all participants. A final data sweep prior to the final analysis will collect information regarding death and any additional anticancer treatment received beyond progression.

Follow-up imaging with MRI will include T2, fluid-attenuated inversion recovery, diffusion-weighted imaging, T1 pre-gadolinium and T1 post-gadolinium sequences. In addition, patients randomised to re-irradiation will also undergo T2* Dynamic Susceptibility Contrast (DSC) perfusion as part of follow-up imaging to assist in determination of the presence of radionecrosis.

Adverse events (AEs), serious adverse events (SAEs), adverse reaction (ARs), serious adverse reactions (SARs)

**Table 2** Trial assessment schedules

| | Early assessments | | | | Follow-up assessment (measured post-start of treatment) | | | | | | | | | |
|---|---|---|---|---|---|---|---|---|---|---|---|---|---|---|
| | Eligibility | Baseline | Pre-treat RT | Pre-treat chemo | 0 weeks | 1 week | 6 weeks | 12 weeks | 18 weeks | 24 weeks | 30 weeks | 36 weeks | 42 weeks | 48 weeks |
| Informed consent | X | | | | | | | | | | | | | |
| Clinical assessment | X | X | | | X | X* | X | X | X | X | X | X | X | X |
| Performance status (Karnofsky) | X | X | | | X | X* | X | X | X | X | X | X | X | X |
| Blood† | X | | | X | X | | X | X | X | X | X | | | |
| Pregnancy screening‡ | X | | X | | X | | X | X | X | X | X | | | |
| Re-irradiation | | | | | X | X | | | | | | | | |
| Chemotherapy | | | | | X | | X | X | X | X | X | | | |
| MRI (RANO criteria) | X | | | | | | | X | | X | | X | | X |
| Toxicity (CTCAE) | | X | | | X | X* | X | X | X | X | X | X | X | X |
| HRQoL | | X | | | | | X | X | X | X | | X | X | X |

*Re-irradiation arm patients only.
†Blood only for eligibility for re-irradiation arm, with each cycle in chemotherapy arm.
‡Only at pretreatment for re-irradiation arm, with every cycle in chemotherapy arm.
CTCAE, Common Terminology Criteria for Adverse Events; HRQoL, health-related quality of life; RANO, Response Assessment in Neuro-Oncology.

and suspected unexpected serious adverse reactions (SUSARs) will all be collected from randomisation until the last participant follow-up or disease progression. Beyond this period only, SARs and SUSARs related to the trial treatment (and not GBM progression) will be reported if the investigator becomes aware of them until the end of trial notification. All SAEs/SARs and SUSARs related to investigation medicinal products will be reported to the Medicines and Healthcare products Regulatory Agency (MHRA) and sponsor. All radiotherapy-related SAEs/SARs and SUSARs will be reported to the Research Ethics Committee and sponsor.

### Statistical analysis

The statistical analysis will be conducted by University of Leeds CTRU. A full statistical analysis plan has been developed and finalised.

The trial analysis will be conducted on a modified intention to treat (MITT) population.[34] MITT is defined as all participants randomised that have received at least one dose of trial treatment, analysed according to the treatment arm they were randomised to (regardless of ineligibility, non-compliance or withdrawal).

OS rates at 9 months, that is, the number and proportion of participants alive at 9 months post-start of treatment, will be calculated for the re-irradiation arm with 90% (corresponding to a one-sided 5% significance level used in the design) and 95% CIs.

Based on the three-outcome design, the cut-off values and conclusions for the statistical test for the primary analysis of efficacy are defined as follows:

Green: If ≥13/33 patients are alive, re-irradiation demonstrates sufficient efficacy to warrant larger-scale evaluation. This is based on a 9-month OS >45% (ie, competitive with the 'best' outcomes from prospective studies using nitrosourea-based chemotherapy).[11 14 35]

Amber: If 11–12/33 patients are alive the decision to take re-irradiation forward will be uncertain and will be based on HRQoL and toxicity at 3 months (ie, based on 9-month OS 25%–45%; ie, similar to outcomes from other prospective nitrosourea-based trials).[13 16 17 36]

Red: If ≤10/33 patients are alive re-irradiation will not be considered worthy of further investigation, based on 9-month OS <25%.

The chemotherapy arm is included for calibration and will be used for validation of outcomes in the re-irradiation group with respect to previous series, and validation of the assumptions used to inform the statistical cut-offs.

Summary statistics will be calculated for all domains of the EORTC QLQ-C30 and QLQ-BN20 and the overall summary score calculated, presented overall and by treatment arm for each of the follow-up time points. The difference in scores between treatment arms will be presented with corresponding 90% and 95% CIs.

Summary statistics will be presented for dexamethasone requirement, anti-epileptic drug requirement, treatment compliance and response assessment; including the proportion of participants within each clinical response

status. The maximum CTCAE grade of toxicities experienced by participants will be summarised for each CTCAE term reported, by treatment arm, for the acute and late toxicities.

The assessment of PFS and OS will be based on a time-to-event analysis and presented using Kaplan-Meier curves. Median PFS and OS will be presented with 95% CIs for each arms. There will be no formal comparison between arms, however, the unadjusted OS HR for the re-irradiation arm vs the chemotherapy arm will be presented.

### Qualitative interviews

Qualitative exploration of HRQoL associated with brain tumour treatments will allow more in-depth evaluation of what matters most to patients and informal caregivers at the time of GBM recurrence; their perception of HRQoL before and during trial treatment; and how they experience both the treatment they receive and trial participation. Interviews will be recorded, transcribed and thematically analysed.[37] A subsample of trial participants will be asked to partake in this interview, around 15 patients and their informal caregivers. We will use a maximum variation sample, to choose patients (and their informal caregivers) that differ in age, sex and previous/current treatment(s) received. By using a varied sample, we aim to highlight important shared patterns that cut across cases and also discover unique or diverse variations from different participants.[38]

### Trial organisation

University of Leeds CTRU will manage all trial and data co-ordination according to the unit's well established standard operating procedures. All trial organisation will be conducted in-line with the principles of Good Clinical Practice.

### Data collection and management

The BRIOChe trial data collection has been designed for remote data entry. However, all safety data will be recorded on paper case report forms and patient-reported outcome measures on the relevant paper questionnaires, sent to and entered at CTRU. Participating centres will record and complete patient data on a trial specific database. The database has instantaneous validation checking. In addition, data management validation reports for missing and discrepant data.

All trial data is stored confidentially in a secure location at University of Leeds CTRU. Access to trial data will be restricted to CTRU staff and study team only, prior to the analysis and release of the trial results. On completion of the trial and publishing of results, researchers can request access to data by contacting members of the BRIOChe Trial Management Group (TMG) or CTRU. All data will be stored and archived for a minimum of 25 years after trial completion.

### Quality assurance

The radiotherapy quality assurance (QA) programme will be implemented by the RTTQA group to ensure treatment is planned and delivered according to the trial protocol. RTTQA includes both benchmarking contouring and planning cases. A summary of the RTTQA requirements are provided in the BRIOChe radiotherapy outlining, planning, treatment delivery and QA guidelines. All sites must complete essential documentation and the site initiation process before a site can be activated. The site initiation will be an electronic process including an audio-visual recording. The video-recorded site initiation presentation slides can be used for future training and reference at participating sites.

### Trial monitoring

The BRIOChe trial will be monitored by the specific project delivery group at CTRU and the multi-disciplinary TMG. An independent Data Monitoring and Ethics Committee (DMEC) will be responsible for the safety and integrity of the patients and study, monitoring unblinded interim data. The DMEC will be formed of two independent clinical oncologists and one independent statistician, meeting at least annually, in addition to receiving 6-monthly safety reports. The DMEC members will advise the Trial Steering Committee (TSC) on any trial or safety concerns. The TSC have definitive control over the continuation of the trial. A DMEC charter has been developed including the roles and responsibilities, and communication processes.

### Patient and public involvement

Patient and public involvement (PPI) has been an instrumental part of the development and conduct of the BRIOChe trial. The trial design was discussed with and presented to PPI representatives. The trial question was felt to be of importance and the treatments and assessment schedule were considered acceptable.

The TMG includes a key patient representative, who provides ongoing and regular advice regarding trial conduct from the patient perspective.

### Imaging substudy

An imaging substudy is planned between two of the recruiting centres (Leeds Teaching Hospitals NHS Trust and University College London Hospitals NHS Foundation Trust), to investigate the impact of re-irradiation and chemotherapy treatments on cerebral vascular changes and to improve understanding as to how these changes might be associated with both tumour response and normal tissue damage. The imaging substudy is estimated to recruit 12–16 patients from the full BRIOChe trial. Substudy participants who have been randomised to re-irradiation will have arterial spin labelling (ASL) and T2 DSC sequences added to the dedicated radiotherapy planning MRI. In addition, these participants will have ASL added to their routine follow-up MRI scans (T2 DSC already included), to provide 'before' and 'after' re-irradiation information. Those substudy participants randomised to chemotherapy will also have ASL and T2 DSC sequences added to their routine follow-up MRI to

allow comparison between chemotherapy-treated and re-irradiated participants.

## Ethics and dissemination

The trial obtained ethical approval from the Office for Research Ethics Committees Northern Ireland (ORECNI) (reference no. 20/NI/0070) and is registered in the ISRCTN registry (registration number 16052954). The trial is currently adhering to protocol version 4.0 (7/12/2022). Any protocol amendments will be submitted to MHRA and REC, all changes will be communicated with local sites.

There is no formal interim analysis. The final trial data will be analysed and reported approximately 1 year after the final participant is recruited. The final trial result manuscript will be written in accordance to ICMJE guidelines and submitted for publication to peer-reviewed journals.

## DISCUSSION

Despite the poor outcomes and high clinical need, there is a lack of active clinical trials for patients with recurrent GBM in the UK and no gold standard treatment at recurrence exists. Chemotherapy is often used in the UK, but outcomes are disappointing. Re-irradiation is recognised as an alternative treatment option and is listed as such in European and American treatment guidelines for patients with recurrent GBM, and in the current National Institute for Health and Care Excellence guidelines for brain tumours.[39–41] Despite this, re-irradiation is less commonly used in the UK compared with other parts of the world. Re-irradiation offers a shorter course of treatment compared with chemotherapy and has a different toxicity profile, and so may be preferable for some patients, although the risks and benefits need further clarification. The BRIOChe trial aims to be a pragmatic, clinically relevant trial that evaluates re-irradiation as an alternative treatment option to chemotherapy for patients with recurrent GBM. While OS is the primary end point, HRQoL is a key secondary endpoint.

The trial design and treatment, specified in BRIOChe, intentionally reflects routine practice. Centres are able to use their usual lomustine-based chemotherapy regimen(s) thus reflecting the variety in practice across the UK. The frequently used and relatively well-tolerated hypofractionated schedule, 35 Gy in 10 fractions, has been adopted for BRIOChe re-irradiation regimen.[27–30]

No OAR constraints have been formally validated for GBM re-irradiation. Most GBM recurrences occur within the previous high dose region and so it is often necessary to assume that a degree of OAR recovery has occurred following the original course of radiotherapy if a meaningful dose is to be delivered to the re-irradiation target. As such, BRIOChe provides 'radiobiological' optimal constraints, allowing for 25% recovery from the original radiotherapy, based on limited evidence regarding post-irradiation recovery in neural structures.[42–44] In addition, where optimal constraints limit re-irradiation delivery, the more lenient mandatory constraints have been based on the RTOG1205 trial.[45] Encouragingly since the opening of the BRIOChe trial, results from the RTOG1205 trial have been published indicating that re-irradiation, 35 Gy in 10 fractions, as used in BRIOChe, in combination with bevacizumab, results in a median OS of 10.1 months.[25] In addition, re-irradiation was well tolerated.[25] However, re-irradiation alone was not investigated in RTOG1205, BRIOChe will investigate this further.

The eligibility criteria volume requirements consider whether the lesion is single or multi-focal. Size and volume limitations were included to limit the amount of normal brain re-irradiation in an effort to keep the risk of radionecrosis low. This approach was preferred over using 'diameter only' measurement as this could potentially exclude long but narrow recurrences, where the total volume was low.

The minimal acceptable interval between completion of original radiotherapy and commencement of re-irradiation is also unknown and this interval varies among existing studies. We have adopted a minimum interval of 6 months from completion of first radiotherapy in BRIOChe, which is in-keeping with current practices in other countries[46] and international trials using the same re-irradiation schedule.[31 32] An interval of at least 6 months also allows some time for OAR repair, should repair occur, and reduces the risk of mistaking pseudo-progression for true progression, which typically occurs within 3 months of radiotherapy.[47]

It is acknowledged as a limitation that the trial design and decision criteria are based on available historical data at the time of development. The referenced studies of nitrosourea-based chemotherapy required intervals between recurrence to treatment that are shorter than the 6 months required in order to enter BRIOChe, potentially selecting some patients that come from a slightly better prognosis patient group. The trial has taken a pragmatic approach given the paucity of data with the exact criteria for those treated with systemic therapy.

While the patient numbers involved will not permit a direct comparison of survival outcomes in each arm, the BRIOChe trial aims to demonstrate that re-irradiation is a possible alternative treatment option for patients with recurrent GBM, including from the patient perspective. This in turn will allow the development of future novel approaches, including potential treatment intensification strategies using re-irradiation in combination with novel agents such as immunotherapies and radiosensitisers, to be explored in large-scale platform studies.

The BRIOChe study will provide valuable data for patients with recurrent GBM treated with re-irradiation and nitrosourea-based chemotherapy, forming a

first step in improving treatment for this poor prognosis patient group.

**Author affiliations**
¹Clinical Trials Research Unit, Leeds Institute of Clinical Trials Research, University of Leeds, Leeds, UK
²Leeds Institute of Medical Research at St James's, University of Leeds, Leeds, UK
³Leeds Institute of Health Sciences, University of Leeds, Leeds, UK
⁴Guy's and St Thomas' Hospitals NHS Trust, London, UK
⁵National Radiotherapy Trials QA (RTTQA) Group, Mount Vernon Cancer Centre, Northwood, UK
⁶Brainstrust, Cowes, UK
⁷Department of Radiology, Leeds General Infirmary, Leeds, UK
⁸Department of Medical Physics, Leeds Cancer Centre, Leeds, UK
⁹Pharmacy, Leeds Teaching Hospitals NHS Trust, Leeds, UK
¹⁰Department of Clinical Oncology, Leeds Cancer Centre, Leeds, UK

**Contributors** Contributors to the conception, trial design and ongoing conduct: LM, FWB, SS, EMH, SN, JW, SRB, OA-S, HBa, HBu, SC, SF, JH, JL, AS, CP, FS, DS-M. Development of the protocol, trial documents and patient information sheet: EMH, LM, SRB, SN, FWB, CP, SC, JL. Writing manuscript: EMH, LM. Reviewed manuscript: all authors. All authors read and approved the final manuscript.

**Funding** BRIOChe is supported by The Jon Moulton Charity Trust (NA), and by Yorkshire Cancer Research (YCR), Centre for Early Phase Clinical Trials (L375PA). The imaging substudy is supported by Cancer Research UK funding for the Leeds Radiotherapy Research Centre of Excellence (RadNet; C19942/A28832). The study is coordinated by University of Leeds CTRU and sponsored by University of Leeds (GA18/118305). For enquires, trial contact email: BRIOCHE@leeds.ac. uk. The sponsor and funders have had no influence on the BRIOChe trial study design, collection, management, analysis and interpretation of data; and decision for publication. The sponsor and funder will review all manuscripts prior to publication.

**Competing interests** FS reports grants from Cancer Research UK, during the conduct of the study. DS-M reports grants from Jon Moulton Charitable Foundation, grants from Yorkshire Cancer Research, during the conduct of the study. LM reports grants from Jon Moulton Charity Trust, grants from Cancer Research UK, during the conduct of the study; grants from Yorkshire Cancer Research, outside the submitted work. SRB reports grants from Jon Moulton Charitable Foundation, grants from Yorkshire Cancer Research, during the conduct of the study. SS reports advisory board activity for Bayer, CeCaVa Laboratory Research project support, Apollomics.

**Patient and public involvement** Patients and/or the public were involved in the design, or conduct, or reporting, or dissemination plans of this research. Refer to the Methods section for further details.

**Patient consent for publication** Not applicable.

**Provenance and peer review** Not commissioned; externally peer reviewed.

**ORCID iDs**
Eleanor M Hudson http://orcid.org/0000-0001-8758-7163
Sarah R Brown http://orcid.org/0000-0002-7975-7537

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
