## [Reviewer comments · BMJ Open]

ARTICLE DETAILS

TITLE (PROVISIONAL)	Brain Re-Irradiation Or Chemotherapy: a phase II randomised trial of re-irradiation and chemotherapy in patients with recurrent glioblastoma (BRIOChe): Protocol for a multi-centre open label randomised trial
AUTHORS	Hudson, Eleanor; Noutch, Samantha; Webster, Joanne; Brown, Sarah; Boele, Florian; Al-Salihi, Omar; Baines, Helen; Bulbeck, Helen; Currie, Stuart; Fernandez, Sharon; Hughes, Jane; Lilley, John; Smith, Alexandra; Parbutt, Catherine; Slevin, Finbar; Short, Susan; Sebag-Montefiore, David; Murray, Louise

VERSION 1 – REVIEW

REVIEWER	Brada, Michael University of Liverpool
REVIEW RETURNED	01-Sep-2023

GENERAL COMMENTS	This is a well written protocol though has some issues which need correcting before the trial is launched and made available in the public domain (in BMJ Open) 1. The statistical considerations in the trial design which will guide the authors in their decision whether to proceed with a further trial is based on studies where patients were recruited 3 months rather than 6 months after completion of RT (specified in their protocol). As BRIOChe will include more favourable group of patients with longer progression free interval the cut off values (306-17) need adjusting accordingly.2. Data on safety of reirradiation of which Ang et al is likely the most reliable is only valid for first treatment to below tolerance (i.e. not including standard GBM RT schedule (60Gy in 30f) which is to and not below tolerance.3. There is some confusion in the trial protocol whether this will be 1:1 (159-167) or 2:1 (211-220) randomisation. The authors do not report the published survival outcome of the only randomised study of reirradiation versus systemic therapy (Tsien et al 2023) which demonstrate no survival benefit of adding reirradiation (while to Bevacizumab, randomised trials of Bevacizumab against alkylating agents show similar efficacy or lack of efficacy). The BRIOChe study is based on the assumption that reirradiation is likely to be more effective. As the only randomised trial evidence suggests that it is unlikely to be the case, it is not justified and ethically acceptable to use a 2:1 rather than 1:1 ratio. It is also recommended the publication of Tsien et al 2023 is included in the introduction and the discussion.
---

	4. The selection criteria include radiotherapy volume requirements which are not available until radiotherapy is planned. The authors do not explain how these are likely to be obtained prior to randomisation (before any radiotherapy preparation/planning is done). The trial protocol does not specify how this is to be obtained to ensure eligibility and whether this will impact on accrual and a potential loss of patients after randomisation. 5. In the Discussion section the authors deviate from their intentions and state “we hope to demonstrate that re-irradiation is an effective and acceptable treatment option for patients with recurrent GBM ...” (467-8). They should refrain from misleading the public on stating what the study will be unable to show particularly as in the protocol they accept the BRIOChe study cannot demonstrate that reirradiation is “effective”. Tsien, C. I., S. L. Pugh, A. P. Dicker, J. J. Raizer, M. M. Matuszak, E. C. Lallana, J. Huang, O. Algan, N. Deb, L. Portelance, J. L. Villano, J. T. Hamm, K. S. Oh, A. N. Ali, M. M. Kim, S. M. Lindhorst and M. P. Mehta (2023). "NRG Oncology/RTOG1205: A Randomized Phase II Trial of Concurrent Bevacizumab and Reirradiation Versus Bevacizumab Alone as Treatment for Recurrent Glioblastoma." J Clin Oncol 41(6): 1285-1295.
--	---

REVIEWER	Eisenstat, David University of Alberta, Oncology
REVIEW RETURNED	29-Sep-2023

GENERAL COMMENTS	Hudson and colleagues present the protocol paper for the BRIOChe phase II, multi-centre, open label, randomised trial for adults with recurrent glioblastoma (GBM). No major concerns are identified. 1. Specific concerns a. The justification for use of the specific re-irradiation treatment schedule (35 Gy in 10 daily fractions) should be explicitly stated earlier in the manuscript. b. It is not entirely clear why there is no planned formal statistical comparison between treatment arms for the decision-making primary analysis. Could some form of assessment for non-inferiority be contemplated? Refer to the Discussion (page 24, line 466), wherein the authors emphasize that this study is not designed to establish superiority of re-irradiation to chemotherapy. c. It is unclear whether cross-over from chemotherapy to re-irradiation or vice versa is permitted at second progression. It is expected that for some patients, this will likely take place. How will this be assessed by the investigators? d. How was 9 months status post commencement of treatment justified as the timepoint to assess the primary objective of this study? e. A key exclusion criteria is the presence of either IDH1/IDH2 mutations. Add "IDH wild-type" to the abstract to further specify the GBM patients to be enrolled on this study. f. For the imaging sub-study (page 21, lines 397-410), how was the number of patients (16) determined? How confident are the study investigators that the proposed number of patients recruited to this sub-study will identify changes both within and across the two
--

	treatment groups (chemotherapy or re-irradiation)? g. The randomisation assignment of 2:1 was changed from the initial 1:1. However, this change to the study design was introduced much later in the protocol manuscript. This may be confusing to the reader who initially views the Figure for the study scheme. 2. Minor concerns a. Page 7, line 120: Add a semi-colon after "recurrence". b. Page 8, line 152: Place a semi-colon after "survival" and add "it is" after "rather" and prior to "to". c. Page 13, line 207: Clarify methylation status as "MGMT promoter methylation status". d. Page 13, line 213: Replace "were" with "are". e. Page 15, lines 270 and 272: Capitalize "Table". f. Page 20, line 382: Change to "oncologists".
--	--

VERSION 1 – AUTHOR RESPONSE

Reviewer: 1

Comments to the Author:

This is a well written protocol though has some issues which need correcting before the trial is launched and made available in the public domain (in BMJ Open)
We thank the author for their positive comment.

1. The statistical considerations in the trial design which will guide the authors in their decision whether to proceed with a further trial is based on studies where patients were recruited 3 months rather than 6 months after completion of RT (specified in their protocol). As BRIOChe will include more favourable group of patients with longer progression free interval the cut off values (306-17) need adjusting accordingly.

The assumed estimates for overall survival of the control arm were estimated based on available data (e.g. Taal et al., 2014; Wick et al., 2017; Batchelor et al., 2013). We acknowledge that several of these trials only required a 3-month interval from radiotherapy to starting chemotherapy and so potentially the BRIOChe population may be more favourable, given the fact the minimum required disease-free survival is at least 6-months. That said, the median time to Lomustine in the Lomustine arm of BELOB was around 10 months. We opted to be pragmatic and use the similar values to this study combined with reference to others. The trial was designed to include chemotherapy patients as a calibration arm for benchmarking outcomes. This inclusion of a calibration arm allows us to gain insights into the chemotherapy survival rates and other relevant data within this specific trial setting and patient population. Based on the design of the trial it would not be suitable to adjust values on observed outcomes. However, we will interpret the final data and results with consideration for both the calibration arm and relevant external data.

2. Data on safety of reirradiation of which Ang et al is likely the most reliable is only valid for first treatment to below tolerance (i.e. not including standard GBM RT schedule (60Gy in 30f) which is to and not below tolerance.

Little is known about the best approach to GBM re-irradiation in terms of OAR dose constraints. Of those who already practiced GBM re-irradiation before the BRIOChe trial, in the UK and beyond, constraints as per RTOG1205 appeared to be the most established. Indeed, this approach is also being used in another trial of re-irradiation that is open in the UK (AZD1390). We therefore largely

adopted these as our mandatory constraints. However, these constraints are 'flat' and don't really consider previous dose. As such, we adopted a more 'dose remaining' approach for our optimal (more conservative) constraints. This requires an assumption of 25% repair to allow somewhat meaningful re-irradiation dose delivery. The RTOG1205 trial has been reassuring in terms of acceptable toxicity levels (accepting in this case bevacizumab was also administered).

3. There is some confusion in the trial protocol whether this will be 1:1 (159-167) or 2:1 (211-220) randomisation. The authors do not report the published survival outcome of the only randomised study of reirradiation versus systemic therapy (Tsien et al 2023) which demonstrate no survival benefit of adding reirradiation (while to Bevacizumab, randomised trials of Bevacizumab against alkylating agents show similar efficacy or lack of efficacy). The BRIOChe study is based on the assumption that reirradiation is likely to be more effective. As the only randomised trial evidence suggests that it is unlikely to be the case, it is not justified and ethically acceptable to use a 2:1 rather than 1:1 ratio. It is also recommended the publication of Tsien et al 2023 is included in the introduction and the discussion.

The BRIOChe trial opened to recruitment in June 2021 (211) and started with a 1:1 randomisation allocation ratio. There have been significant challenges with slow recruitment for the trial mid/post pandemic in a recurrent GBM setting, and various strategies for reducing the number of patients required in the trial have been discussed. The proposed change in ratio (2:1) was favoured over a one arm continuation (re-irradiation only) in order to maintain the randomisation element. We would like to highlight that the sample size is based on the single arm design requiring 35 re-irradiation patients; the change to this allocation ratio does not increase the number of patients exposed to re-irradiation. The decision was discussed with trial management group, data monitoring committee, and trial steering committees, and agreed to be an appropriate trial design change.

While the results of this combination treatment (Bevacizumab + re-irradiation) do not show any additional benefit over Bevacizumab alone in overall survival, there would be more of an ethical question if we were adding re-irradiation to our PCV chemotherapy regimen. However, BRIOChe is exploring re-irradiation as an alternative isolated treatment, where there may be other benefits to treatment. Re-irradiation was considered safe in terms of toxicity (Tsien et al., 2023). The BRIOChe study does not aim to directly compare re-irradiation with chemotherapy but instead to obtain efficacy and toxicity data to justify the use of re-irradiation as an acceptable treatment option in this population.

We agree the data from Tsien et al., is valuable has now been specifically highlighted to the background information (144-149) and the discussion (472-476).

4. The selection criteria include radiotherapy volume requirements which are not available until radiotherapy is planned. The authors do not explain how these are likely to be obtained prior to randomisation (before any radiotherapy preparation/planning is done). The trial protocol does not specify how this is to be obtained to ensure eligibility and whether this will impact on accrual and a potential loss of patients after randomisation

The number, size and volume of recurrent disease, and therefore eligibility for BRIOChe, will be determined from the diagnostic MRI scan on which the recurrence is identified. The protocol provides instructions on estimating GTV volume based on the volume of an ellipsoid. In addition, in case of significant progression at the time of reRT planning, the RT guidelines state: Patients in whom the GTV has progressed between the baseline MRI and the MRI used for fusion to the planning CT scan, such that the actual GTV volume has progressed to >75cm³ and/or the volume is more than double what it was on the baseline MRI, should be discussed with the RTTQA group before proceeding with re-irradiation.

5. In the Discussion section the authors deviate from their intentions and state "we hope to

demonstrate that re-irradiation is an effective and acceptable treatment option for patients with recurrent GBM ...” (467-8). They should refrain from misleading the public on stating what the study will be unable to show particularly as in the protocol they accept the BRIOChe study cannot demonstrate that reirradiation is “effective”.

We accept the term ‘effective’ in this setting is less appropriate, given the patient population and trial design. We have therefore removed the term from the sentence, now reading ‘the BRIOChe trial aims to demonstrate that re-irradiation is an acceptable alternative treatment option for patients with recurrent GBM’ (488-490).

Reviewer: 2

Comments to the Author:

Hudson and colleagues present the protocol paper for the BRIOChe phase II, multi-centre, open label, randomised trial for adults with recurrent glioblastoma (GBM).

No major concerns are identified.

1. Specific concerns

a. The justification for use of the specific re-irradiation treatment schedule (35 Gy in 10 daily fractions) should be explicitly stated earlier in the manuscript.

We have now added specific reference to the treatment schedule earlier in the manuscript, specifically referenced background>> rational for the study (158).

b. It is not entirely clear why there is no planned formal statistical comparison between treatment arms for the decision-making primary analysis. Could some form of assessment for non-inferiority be contemplated? Refer to the Discussion (page 24, line 466), wherein the authors emphasize that this study is not designed to establish superiority of re-irradiation to chemotherapy.

To formally statistically compares the two treatments, requires a larger number of patients and as such would be prohibitive in terms of patient availability and feasibility of the trial regarding timelines(162-163). For instance, a 1-sided test with 0.1 alpha, assuming the proportion of patients alive at 9 months is 0.25 in control arm and 0.45 in experimental, would provide only 56% power in a sample of 35 patients in each arm (adjusted for loss). With the adaption of a 2:1 part way through, we will have a smaller number of patients in the chemotherapy arm (estimated 20) and so even less power for the formal direct comparison. In a non-inferiority setting where the acceptable difference is most likely smaller, even more patients would be required, meaning an infeasible trial design for BRIOChe patient population and trials funding.

The BRIOChe study will explore efficacy, toxicity, and quality of life data for GBM patients in a randomised trial. The trial aims to provide sufficient evidence to support the use of reirradiation as an acceptable alternative treatment option for this patient population. The paper text has been updated to add clarity to this aim (488-490). Within this rare population, it was critical to utilise a pragmatic approach to answer question in timely manner, therefore this non-comparative design was deemed appropriate.

c. It is unclear whether cross-over from chemotherapy to re-irradiation or vice versa is permitted at second progression. It is expected that for some patients, this will likely take place. How will this be assessed by the investigators?

We detail in the manuscript: at the point of disease progression, patients may receive the alternative treatment according to local policy (281-282). It is therefore possible for cross over of treatments post progression. Data will be collected from all participants, until progression or death (287-2288) and as such the interpretation of endpoints will be carefully considered. The protocol includes collection of planned treatment after progression, in addition to a final data sweep prior to final analysis, collecting information regarding any additional anti-cancer treatments that the patient received beyond progression, additional information now added (289-290). Given this is a phase II trial with a single arm primary endpoint design, focused on raw numbers of patients alive, there will be no adjustment for treatment received post-progression. However, any data received on this will be summarised by arm to aid in interpretation and understand the treatment received.

d. How was 9 months status post commencement of treatment justified as the timepoint to assess the primary objective of this study?

The 9 months overall survival primary endpoint was selected from the upper estimate of median overall survival from those treated with chemotherapy for recurrent GBM, approximately 6-9 months [12,16-23]. The endpoint also aligns with large existing studies of systemic agents for recurrent GBM (i.e. BELOB trial).

e. A key exclusion criteria is the presence of either IDH1/IDH2 mutations. Add "IDH wild-type" to the abstract to further specify the GBM patients to be enrolled on this study.

Addition of IDH wild-type has now been added to the abstract (51).

f. For the imaging sub-study (page 21, lines 397-410), how was the number of patients (16) determined? How confident are the study investigators that the proposed number of patients recruited to this sub-study will identify changes both within and across the two treatment groups (chemotherapy or re-irradiation)?

The imaging sub-study is designed to be hypothesis generating and no planned direct comparisons between groups. It is dependent on the availability of specific MRI sequences, and therefore the sub-study will only be performed at 2 specialist centres. Recruitment of 16 participants to the sub-study was estimated based on anticipated recruitment rates at these 2 centres to the main BRIOChe study. Due to the MHRA delay in reviewing protocols during 2023, the implementation of this sub-study amendment was significantly delayed and therefore likely to affect recruitment of the 16 patients. This has been rephrased as an expected number of patients and given a range of 12-16 (419-420).

g. The randomisation assignment of 2:1 was changed from the initial 1:1. However, this change to the study design was introduced much later in the protocol manuscript. This may be confusing to the reader who initially views the Figure for the study scheme.

The study scheme now has an additional footnote to explain that this ratio was changing during the recruitment phase. Additional clarification on this change has also been added higher up in text (174-179).

2. Minor concerns

a. Page 7, line 120: Add a semi-colon after "recurrence". Updated

b. Page 8, line 152: Place a semi-colon after "survival" and add "it is" after "rather" and prior to "to". updated

c. Page 13, line 207: Clarify methylation status as "MGMT promoter methylation status". Updated

d. Page 13, line 213: Replace "were" with "are". Updated

- e. Page 15, lines 270 and 272: Capitalize "Table". Updated
- f. Page 20, line 382: Change to "oncologists". Updated

VERSION 2 – REVIEW

REVIEWER	Brada, Michael University of Liverpool
REVIEW RETURNED	21-Nov-2023

GENERAL COMMENTS	Response to the comments. 1. The statistical considerations in the trial design which will guide the authors in their decision whether to proceed with a further trial is based on studies where patients were recruited 3 months rather than 6 months after completion of RT (specified in their protocol). As BRIOChe will include more favourable group of patients with longer progression free interval the cut off values (306-17) need adjusting accordingly. It is noted that the authors accept the above comments particularly as the two largest trials [Phase II randomised BELOB trial (Taal et al 2014) and Phase III EORTC trial (Wick et al 2017)] on which the 9 month survival values which determine the continuation of the trial into phase III will be based accrued patients with minimum 3 months post radiotherapy rather than the 6 months in BRIOChe. The cut off values in the 3-outcome design therefore need adjusting. The reasoning given by the authors for not doing so is difficult to understand (and likely not appropriate) particularly as this is such a critical aspect of the assessment of their hypothesis. The proposed 2:1 design means that “benchmarking/calibration” to chemotherapy is relegated to only 20 chemotherapy patients. It means it is pretty much just a single arm Phase II study where the comparative values have to be as accurate as possible. In this setting it also remains unclear why a 6 months disease free period is chosen in this study as the NRG Oncology/RTOG1205 trial of reirradiation (Tsien et al 2022) which the authors accept as showing it is reasonably safe and well tolerated. It is accepted the outcome of this study was only known well after the trial was designed and started accruing. Nevertheless the requirement for the long disease free interval may be one of the reasons for poor accrual so far. 3. Changing 1:1 to 2:1 randomisation Assuming chemotherapy and reirradiation are equally effective (or ineffective) and this is likely the investigators view (now the reference to efficacy has been removed), the change would indeed not be unethical. If poor accrual is the main stumbling block to completing the trial and 2:1 randomisation largely renders to “comparative” chemotherapy arm largely redundant the alternative would have been doing away with randomisation. While it is accepted that 2:1 would therefore not be unethical the change undermines the rationale of the original design.
--

	4. Radiotherapy volume requirements It would be of value if this information was included in the manuscript, as it is presumably aimed at clinical oncologists in other centres to aid accrual.
--	---

REVIEWER	Eisenstat, David University of Alberta, Oncology
REVIEW RETURNED	01-Dec-2023

GENERAL COMMENTS	The author team has substantively addressed the concerns raised through the initial review. The revised protocol manuscript is improved as a result. No further revisions are requested. Two minor typographical errors require attention as follows:  1. Line 121: Add "it" prior to "is usually reserved...". 2. Line 476: Change "FTOG" to "RTOG".
---

VERSION 2 – AUTHOR RESPONSE

Reviewer: 1

Dr. Michael Brada, University of Liverpool

Comments to the Author:

Response to the comments.

“1. The statistical considerations in the trial design which will guide the authors in their decision whether to proceed with a further trial is based on studies where patients were recruited 3 months rather than 6 months after completion of RT (specified in their protocol). As BRIOChe will include more favourable group of patients with longer progression free interval the cut off values (306-17) need adjusting accordingly.

It is noted that the authors accept the above comments particularly as the two largest trials [Phase II randomised BELOB trial (Taal et al 2014) and Phase III EORTC trial (Wick et al 2017)] on which the 9 month survival values which determine the continuation of the trial into phase III will be based accrued patients with minimum 3 months post radiotherapy rather than the 6 months in BRIOChe. The cut off values in the 3-outcome design therefore need adjusting. The reasoning given by the authors for not doing so is difficult to understand (and likely not appropriate) particularly as this is such a critical aspect of the assessment of their hypothesis. The proposed 2:1 design means that “benchmarking/calibration” to chemotherapy is relegated to only 20 chemotherapy patients. It means it is pretty much just a single arm Phase II study where the comparative values have to be as accurate as possible.”

BRIOChe is an ongoing, peer-reviewed trial with pre-determined cut values essential for the study design, directly linked to sample size and operating characteristics (e.g. error rates). We maintain the view that altering the trial design, including estimates, sample size, and errors mid-study, particularly with available outcome data on the primary endpoint, is not appropriate. The trial is overseen by both a data monitoring and trial steering committee, focused on study integrity, data interpretation, and decision guidance.

We have taken a pragmatic approach to defining our historic control data given the paucity of trial data with the exact criteria. While we acknowledge potential differences in population estimates as a limitation, with additional detail provided in the discussion (512-518), we emphasise the importance of having concurrent control arm data for multiple outcomes within the same trial population. Due to the MHRA delay in reviewing protocols during 2023, the protocol amendment and adaption to a 2:1 ratio, it is estimated to be nearer 25 patients treated with chemotherapy. Though the cut values will not change, any potential differences in the population from the original estimates will be thoroughly considered in the final results interpretation, along with concurrent trial data and emerging literature.

“1. In this setting it also remains unclear why a 6 months disease free period is chosen in this study as the NRG Oncology/RTOG1205 trial of reirradiation (Tsien et al 2022) which the authors accept as showing it is reasonably safe and well tolerated. It is accepted the outcome of this study was only known well after the trial was designed and started accruing. Nevertheless the requirement for the long disease free interval may be one of the reasons for poor accrual so far.”

Six months from completion of primary radiotherapy to re-irradiation is generally considered the minimum acceptable interval. This was used in RTOG1205 and the current AZD1390 and RISING trials of re-irradiation. A 6 month interval allows time for normal tissue repair following the first course of radiotherapy and takes patients beyond the window where there would be the highest potential risk of mistaking true progression and pseudo-progression. We acknowledge that this relatively 'longer' interval for eligibility may be one potential explanation for slow accrual, however, remain confident this is the most appropriate length for the BRIOChe trial and would be the minimum accepted interval for re-irradiation in routine clinical practice.

“3. Changing 1:1 to 2:1 randomisation

Assuming chemotherapy and reirradiation are equally effective (or ineffective) and this is likely the investigators view (now the reference to efficacy has been removed), the change would indeed not be unethical. If poor accrual is the main stumbling block to completing the trial and 2:1 randomisation largely renders to “comparative” chemotherapy arm largely redundant the alternative would have been doing away with randomisation. While it is accepted that 2:1 would therefore not be unethical the change undermines the rationale of the original design.”

An advantage of a randomised phase II trial with a calibration arm over a single-arm approach is its ability to uphold the principles of randomisation, reducing the impact of selection bias. Though the 2:1 adaption reduces the number of patients in the calibration arm, we believe it valuable to have concurrent control arm data on multiple outcomes within the same population, rather than relying solely on historical data for interpretation. Despite encountering challenges in recruitment, we maintain this randomisation element preserves the study's original design and rationale, distinguishing it from single-arm, non-randomised alternatives, and providing much needed evidence in the absence of selection bias. Again, this was supported and favoured by the independent trial committees.

“4. Radiotherapy volume requirements

It would be of value if this information was included in the manuscript, as it is presumably aimed at clinical oncologists in other centres to aid accrual.”

We have provided additional information regarding eligibility assessment and radiotherapy volumes to the manuscript (211-219).

Reviewer: 2

Dr. David Eisenstat, University of Alberta

Comments to the Author:

“The author team has substantively addressed the concerns raised through the initial review. The revised protocol manuscript is improved as a result. No further revisions are requested.

Two minor typographical errors require attention as follows:

1. Line 121: Add "it" prior to "is usually reserved...".
2. Line 476: Change "FTOG" to "RTOG".

VERSION 3 – REVIEW

REVIEWER	Brada, Michael University of Liverpool
REVIEW RETURNED	26-Jan-2024

GENERAL COMMENTS	The authors did not adjust for the H1 endpoint as suggested (in two reviews). It is currently likely set too low which may inappropriately favour continuing a phase III trial of reirradiation. Nevertheless the overall manuscript is fine except for two issues which require attention. 1. P.8 para 1 of the tracked revised manuscript when referring to RTOG 1205 trial the authors single out “improved progression free survival” with radiotherapy ignoring the principal result which is no difference in survival. This is disingenuous especially as Bevacizumab does not prolong survival in randomised studies in comparison to standard care (and therefore unlikely to be effective) which somewhat inconveniently suggests the same for reirradiation. The results of the trial should be specified in full. 2. P.8 para 3 the authors added a statement “the BRIOChe trial aims to demonstrate re-irradiation as an acceptable alternative treatment option for patients with recurrent GBM”. The trial is not designed to come up with such a conclusion and the additional sentence should be removed.
---

VERSION 3 – AUTHOR RESPONSE

"Reviewer: 1

Dr. Michael Brada, University of Liverpool

Comments to the Author:

The authors did not adjust for the H1 endpoint as suggested (in two reviews). It is currently likely set too low which may inappropriately favour continuing a phase III trial of reirradiation. Nevertheless the overall manuscript is fine except for two issues which require attention."

"1. P.8 para 1 of the tracked revised manuscript when referring to RTOG 1205 trial the authors single out “improved progression free survival” with radiotherapy ignoring the principal result which is no difference in survival. This is disingenuous especially as Bevacizumab does not prolong survival in randomised studies in comparison to standard care (and therefore unlikely to be effective) which somewhat inconveniently suggests the same for reirradiation. The results of the trial should be specified in full."

The following sentence “There was no difference in overall survival between arms, the primary endpoint of the trial, however, progression free survival was improved compared to bevacizumab alone” has been added to the manuscript for further detail.

"2. P.8 para 3 the authors added a statement “the BRIOChe trial aims to demonstrate re-irradiation as an acceptable alternative treatment option for patients with recurrent GBM”. The trial is not designed to come up with such a conclusion and the additional sentence should be removed."

We acknowledge BRIOChe is not a phase III trial, and will not provide definitive evidence on re-irradiation, and we would not wish for this sentence to be interpreted in the way. We have therefore, adapted the sentence to the following: “the BRIOChe trial aims to demonstrate re-irradiation as a possible alternative treatment option for patients with recurrent GBM and inform the design of future research”.